# The Remarkable Properties of Oil-in-Water Zein Protein Microcapsules

**DOI:** 10.3390/molecules31010153

**Published:** 2026-01-01

**Authors:** Alessandra Quarta, Chiara Del Balzo, Francesca Cavalieri, Raffaella Lettieri, Mariano Venanzi

**Affiliations:** Department of Chemical Science and Technologies, University of Rome Tor Vergata, Via della Ricerca Scientifica 1, 00133 Rome, Italy; alessandraquarta030897@gmail.com (A.Q.); chiara.delbalzo22@gmail.com (C.D.B.); francesca.cavalieri@uniroma2.it (F.C.); raffaella.lettieri@uniroma2.it (R.L.)

**Keywords:** core-shell microcapsules, curcumin, protein microcapsule, ultrasound-assisted emulsification, zein protein

## Abstract

Zein (ZP) is the major storage protein of corn (maize). It is safe, biodegradable, edible, and characterized by unique self-assembly properties. These properties were exploited to prepare ZP microcapsules filled with soybean oil (SO) by ultrasound-assisted emulsification of oil-in-water (o/w) dispersions under optimal experimental conditions. The morphology and stability of o/w ZP/SO microcapsules were investigated by optical spectroscopy (electronic circular dichroism and fluorescence) and dynamic light scattering, as well as bright-field, laser confocal fluorescence, and scanning electron microscopies. The results showed that ZP formed a stable protein shell protecting the inner oily phase from diffusion of the confined compounds. It was also found that ZP/SO microcapsules, stored under suitable conditions, could be redissolved in water, maintaining their spherical morphology. Proof-of-principle studies on the inclusion and release of curcumin, a very active anti-inflammatory and nutraceutical substance, from ZP/SO microcapsules under temperature and pH stimuli are also reported.

## 1. Introduction

Encapsulation of active ingredients in microparticles is a green technology with a wide range of applications in food chemistry, cosmetics, and theranostics [1]. Examples of microsystems able to entrap active materials include polymeric nano- and microparticles, core-shell structures, microsponges, and microspheres forming stable colloid emulsions in solution [2,3].

Among the many available techniques, ultrasound-assisted emulsification (UAE) is a simple, low-cost, and effective method for preparing aqueous suspensions of oil-in-water (o/w) microcapsules filled with active ingredients [4,5]. The protein shell prevents the diffusion of the confined compounds out of the inner phase, providing a barrier of nanometric thickness that can protect the embedded molecules from acidic, alkaline, or enzymatic degradation. Moreover, polymeric capsules may respond to changes in pH, temperature, or other external stimuli, enabling the release of entrapped materials under specific experimental conditions [6]. Microcapsules made from proteins have the advantage of being intrinsically biocompatible and biodegradable. Due to these properties, they have been extensively studied for pharmaceutical and nutraceutical applications [7,8].

Zein protein (ZP), the primary storage protein in maize, is FDA-recognized as safe [9]. It is biodegradable, edible, and exhibits unique self-assembly properties, with abundant availability from corn surpluses and bioethanol byproducts [10]. Applications of ZP include use in fibers, adhesives, coatings, ceramics, inks, cosmetics, textiles, and biodegradable plastics [11,12,13,14]. Furthermore, ZP is known to form a robust, hydrophobic, grease-proof coating that can be used in the preparation of biodegradable films and plastics, ensuring resistance against microbial attack [15,16]. However, ZP’s insolubility in water and its poor nutritional quality (negative nitrogen balance) limit its use in human food products, though these properties promote its industrial utilization in food packaging and coating [17].

From a biochemical point of view, ZP is encoded by a large multigene family and constitutes a heterogeneous mixture of polypeptides with different molecular sizes, solubilities, and charges [18]. Four main ZP fractions have been identified, i.e., α (19–22 kDa, 70–80%), β (17 kDa, 10–20%), γ (27 kDa), and δ (10 kDa) ZP. Presently, more than 60 structures are deposited on the ZP UniProt KB database [19], with more than 40 for α-ZP [20]. The low solubility of ZP in water and its prevalent hydrophobic character are primarily attributed to the high content of non-polar amino acids (more than 50%) in the protein primary structure, as well as the amidation of glutamic residues (21–26%) into glutamines and asparagines. However, ZP is soluble in binary solutions consisting of lower aliphatic alcohols and water, such as 50–90% aqueous ethyl alcohol (EtOH) [21].

The secondary structure of ZP has been investigated by electronic circular dichroism (ECD) and FTIR spectroscopy. It has been reported that ZP is characterized by a ~40–60% helical content, complemented by β-sheet (30%) and β-turn (20%) structures [22,23,24], although there is no consensus on the tertiary structure of ZP. Argos et al. [25] proposed a ZP 3D model comprising nine adjacent, topologically antiparallel helices, clustered by glutamine-rich turns or loops The polar and hydrophobic residues distributed along complementary helical surfaces facilitate intra- and intermolecular hydrogen bonding, allowing ZP to be arranged in planes. It has also been shown that ZP is able to self-assemble into mesophases of different morphologies depending on the specific experimental conditions [26,27]. In particular, ZP may lead to the formation of stable nanostructures, such as fibers, nanoparticles, microspheres, and thin films, depending on the experimental conditions [28].

UAE is a well-assessed technique for the synthesis of oil-filled water-soluble microcapsules. Oil emulsification is achieved by the shear stress produced by the collapse of microbubbles under ultrasound pressure through multiple in-phase expansion and contraction steps [4]. Acoustic cavitation of bubbles boosts the disruption of oil droplets, facilitating the formation of stable o/w emulsions [5].

Cavitation effects strongly depend on the applied ultrasound frequency. The amount of energy released by the bubble collapse and the maximum bubble size before collapse (resonance size) are correlated and inversely proportional to the applied frequency. Ultrasound parameters, such as the frequency and the acoustic power, can be tuned to control the number of cavitation events, the microcapsule size, and the surface roughness and structure [4].

Ultrasound has already been used to prepare microcapsules stabilized by a protein network coating bubbles, droplets, or other templates [29]. During the emulsification stage, bubbles or liquid droplets act as a template for the protein shell, which forms a physical barrier at the interface between the inner oily phase and the aqueous solution. Reversible protein coating is achieved by exploiting the hydrophobic effect and hydrogen bonding interactions, while irreversible coating takes place if covalent linkages, such as disulfide bonds, are produced. In the latter case, the formation of covalent links can be facilitated by free radicals generated through high-frequency ultrasonic cavitation [4,5].

During UAE, ZP self-assembles at the interface of the oil droplets dispersed in water, thus forming a tight and effective protective coating for the encapsulated compounds [30]. In a seminal paper, Wang et al. used surface plasmon resonance experiments to show that ZP was able to adsorb to hydrophobic and hydrophilic surfaces with the same affinity [31]. During the protein structural reorganization at the o/w interface, ZP preferentially exposes its hydrophobic surface to the inner oily phase and its hydrophilic surface to the outer water solution, according to the ZP 3D model proposed by Argos et al. [25].

While several studies have investigated ZP/polymer and ZP/surfactant systems as stabilizers in emulsion formulations [32,33,34,35,36], relatively few efforts have been dedicated to the ultrasound-assisted production of pure ZP-coated microcapsules, with alternative methods generally being preferred [30,37,38,39,40].

In this study, the stability and morphology of o/w ZP/soybean oil (SO) microcapsules (MCs), denoted hereafter as ZP/SO MCs, were investigated by optical spectroscopy, dynamic light scattering, and optical and electronic microscopy techniques. Furthermore, proof-of-principle experiments on the inclusion of curcumin in ZP/SO MCs and its release as a result of temperature and pH stimuli will be discussed.

The principal elements of novelty of this study are as follows: (i) the long-term stability of ZP/SO MCs produced under an optimal ZP/SO concentration ratio and sonication conditions; (ii) the capacity to redisperse ZP/SO MCs, stored under suitable conditions, in 100% water without loss of morphology; and (iii) the capacity of ZP/SO MCs to load and release curcumin upon pH and temperature stimuli. These remarkable properties are principally determined by the structural reorganization of the ZP shell at the o/w interface, as indicated by the formation of dityrosine groups through tyrosine–tyrosine linkages and the ZP structural transition from α-helical to antiparallel β-sheet conformations.

## 2. Results

### 2.1. Preparation of Oil-in-Water Zein/Soybean Oil Microcapsules by Ultrasound-Assisted Emulsification

Details on the optimization of the experimental conditions for the preparation of ZP/SO MCs by UAE are briefly reported in the Section 3 (see below) and, more extensively, in the Appendix A. Very briefly, a set of experiments was designed to evaluate the microcapsules’ stability and size distributions by varying several preparation parameters (Appendix A): acoustic power (Appendix A), sonication time (Appendix A), ZP (Appendix A), SO concentrations (Appendix A), and storage conditions (Appendix A). Each system was monitored for a minimum of 5 days to a maximum of one month.

The optimal sonication conditions were therefore fixed as follows: 45 s sonication time, 20 kHz frequency, and 220 W electric power. Under these experimental conditions, minimal oil/water phase separation was detected after 10 days. Specifically, stable ZP/SO MCs were obtained only for ZP concentrations higher than 2 mg/mL. Finally, UAE of the formulation composed of 10 µL SO and 5 mg/mL ZP solutions in 1 mL EtOH/H_2_O 70/30 (*v*/*v*) provided the best results in terms of stability and homogeneous size distribution of ZP/SO MCs.

The formation of ZP/SO MCs could be clearly detected by optical microscopy imaging right after the ultrasound treatment (Figure 1A) and up to 4 days of storage at room temperature (Figure 1D). In Figure 1D, a crown of protein nanocapsules decorating the border of micrometric ZP/SO MCs can be observed, indicating the occurrence of a time-dependent coalescence process. Optical microscopy images of ZP/SO MCs produced by UAE of dispersions of different ZP concentrations (5, 7.5, and 10 mg/mL) and SO amounts (10, 25, and 50 µL) are reported as SM (Appendix A). It was found that the MC number density and stability were highly dependent on the ZP/SO concentration ratio.

For comparison, UAE of an EtOH/H_2_O 70/30 (*v*/*v*) and SO dispersion was carried out under the same sonication conditions, and the results are shown in Appendix A. A rapid separation between the oil and aqueous phases was observed, with no evidence of regularly shaped microstructures.

Interestingly, dried ZP/SO MCs, stored under optimal conditions, could be successfully redispersed in 100% water, maintaining the same morphology, structure, and size distribution, even after 25 days from redissolution (Figure 2). In particular, the size distribution of protein microcapsules was found to be centered between 0.7 and 1.2 μm at all times. We consider this finding a crucial result for future applications of ZP-coated MCs. Fluorescence and circular dichroism experiments (Section 2.3) provided some insights into the reasons for the remarkable stability of ZP/SO MCs in water, despite the low solubility of ZP in aqueous solutions.

### 2.2. Morphological Characterization of o/w ZP/SO Microcapsules

The bright-field optical microscopy image reported in Figure 3 clearly shows the spherical morphology of ZP/SO MCs. The regular shape of the protein microcapsules is a clear indication of the surfactant properties of ZP. We are therefore inclined to identify the dark border of the microcapsules as the ZP shell, positioned at the o/w interface, and attribute the more transparent inner phase to SO.

The stability and morphology of ZP/SO MCs were closely monitored for 5 days. It was found that the protein microcapsules, produced under optimal experimental conditions, remained in a metastable state for several days, despite the insurgence of a time-dependent coalescence process. This phenomenon is clearly visible in the bright-field optical microscopy images of ZP/SO MCs reported in Figure 4, taken 4 days after preparation.

Interestingly, in the inner oily phase of the larger microcapsules shown in Figure 4, nanometric ZP coacervates can be easily recognized. Analysis of the size of ZP/SO MCs provided average diameters in the 9–17 μm range, although microcapsules featuring diameters of about 100 μm could also be seen (Figure 4B).

Nile red (NR), a well-known solvatochromic dye, was included in the oily phase of o/w ZP/SO MCs to monitor the oil partition in the microcapsules prepared by UAE. NR is an excellent dye for the detection of intracellular lipid droplets by fluorescence microscopy because its fluorescence emission, strongly enhanced in a hydrophobic environment, is almost completely quenched in water [41]. Recently, Weissmueller et al. successfully encapsulated NR in the hydrophobic environment of ZP nanocarriers, obtained by flash nanoprecipitation [13].

NR-labeled ZP/SO MCs were therefore observed by optical microscopy, collecting both bright-field and fluorescence images. The microstructures revealed by the two imaging techniques strongly overlapped, indicating that the ZP/SO MCs are filled with SO/NR oil (Appendix A). Clustered ZP/SO/NR MCs produced by coalescence (Appendix A) were clearly visible 3–4 days after the UAE preparation, suggesting that the formation of large microcapsules proceeds through the merging of ZP outer layers and the pooling of the oil content.

To obtain further evidence on their core-shell morphology, ZP/SO MCs were also stained with rhodamine B (RhB). It is well known that RhB can be easily conjugated to ZP, exploiting electrostatic bonds, hydrogen bonds, and van der Waals interactions [42].

ZP/SO MCs embedded with NR in the SO inner phase (Figure 5A) or stained with RhB (Figure 5B) were therefore imaged by confocal laser scanning fluorescence microscopy (CLSFM). The consistency of the red fluorescence emission signals in NR- and RhB-stained samples can be easily seen in Figure 5A,B. Besides the confirmation of the NR localization in the inner oily phase, clear evidence of the ZP(RhB) crown shell at the o/w interface can be observed in the RhB-stained microstructures (Figure 5B), together with the faint background emission of RhB in the solution. A distinct demarcation exists between the clustered microcapsules when SO is stained by NR, and the net borderline denotes the ZP(RhB) fluorescent crown. These results strengthen the idea that ZP/SO MCs are structured as an SO-filled core protected by a ZP outer shell.

Field emission environmental scanning electron microscopy (FE-ESEM) experiments [43,44] were also carried out comparing freshly synthesized ZP/SO MCs with the same sample after 18 days of storage at room temperature (Figure 6). The FE-ESEM image of an EtOH/H_2_O 70/30 (*v*/*v*)/SO emulsion (control sample) is also reported as SM (Appendix A) for comparison. Contrary to what was observed in Appendix A, where microstructures of any morphology could be detected, many ZP/SO microstructures were imaged by FE-ESEM (Figure 6), further evidencing the role of the ZP shell in the stabilization of the oil droplets. However, freshly synthesized ZP/SO MCs showed remarkable morphological differences with respect to 18-day-old samples. In particular, the 18-day-old protein microstructures gave rise to rather elongated droplets, denoting degradation of the protein shell integrity and SO leakage (note the different scales of Figure 5A,B).

### 2.3. Optical Spectroscopy Studies of ZP/SO Microcapsules

UV absorption, fluorescence emission, and ECD studies were performed to obtain an insight into the secondary structure of ZP positioned at the o/w interface of ZP/SO MCs. We found that ZP, dissolved in 70/30 (*v*/*v*) EtOH/H_2_O solution, showed two absorption bands in the near-UV wavelength region, peaking at λ = 280 nm and λ = 320 nm. These electronic transitions can be assigned to the Tyr [45] and dityrosine chromophores, respectively [46] (Appendix A). Dityrosine is a natural component of ZP, produced by environmental stress and protein modification [47], and it has already been detected and quantified in previous studies [48].

ZP fluorescence emission spectra obtained by excitation at λ_ex_ = 283 nm and λ_ex_ = 315 nm are reported in Appendix A. In the figure, an emission maximum at λ_em_ = 310 nm, typical of the tyrosine monomer, and a long-wavelength broad emission extending from 340 to 540 nm, ascribable to dityrosine groups, can be observed [46]. Interestingly, the fluorescence emission spectrum of ZP in ZP/SO MCs (Figure 7) shows a substantial quenching of the Tyr monomer emission and a relatively large contribution of the dityrosine emission. This finding suggests that Tyr–Tyr interactions and the formation of dityrosine crosslinks may contribute to stabilizing the protein shell of ZP/SO MCs.

Important insights into the structural modification of ZP involved in the formation of the ZP/SO MC protein shell are provided by ECD experiments. The ECD curves reported in Figure 8 clearly show that, while ZP in EtOH/H_2_O 70/30 *v*/*v* solution predominantly attained a helical conformation, ZP in ZP/SO MCs underwent a conformational transition, populating a β-sheet secondary structure. Heat-induced structural rearrangements from helical to β-sheet conformations have been found to occur during the formation of peptide aggregates and gels [49]. The role of heat is to promote the disruption of the helical secondary structure, favoring the population of unfolded conformations that finally rearrange in a stable β-sheet layer. In the case of UAE, local heating of ZP represented the driving force promoting the protein structural rearrangement that led to the formation of a stable protein shell at the o/w interface. Together with the fluorescence evidence reported above, the ECD results suggest that the protein shell in ZP/SO MCs is further stabilized by layering interactions between ZP complementary surfaces, attaining a β-sheet conformation. This conclusion is supported by the deconvolution of ECD curves carried out through the BestSel code (Table 1) [50], which confirms that a conformational transition from helical to β-sheet structures is taking place when ZPs in solution are involved in the formation of ZP/SO MCs.

### 2.4. Stability of ZP/SO Microcapsules under Temperature and pH Stimuli

The stability of ZP/SO MCs in the 25–70 °C temperature range and 2–11 pH interval was investigated by Rayleigh light scattering (RLS) and dynamic light scattering (DLS) experiments. As can be observed in Appendix A, the RLS intensity of ZP/SO MCs solutions showed a continuous decrease in the temperature range between 30 °C and 70 °C. This finding could be associated with a drop in the particle number density and dimensions. DLS measurements revealed that the average hydrodynamic radius of ZP/SO MCs strongly decreased with the temperature, varying from 2.0 ± 0.2 μm at 25 °C to 200 nm at 70 °C (Appendix A).

Thermal bursting preferentially occurs for microcapsules characterized by a particle size larger than a “critical dimension”. Resistance to thermal disruption increases as the microcapsule size decreases, following an exponential trend similar to those reported in Appendix A [51]. The results can therefore be explained in terms of the progressive breaking of ZP/SO MCs of larger dimensions as the temperature increases, with only microcapsules of nanometric dimensions able to survive.

A ZP/SO MC size reduction was also observed by RLS measurements at pH 2, while at pH larger than 7.3 (isoelectric point), the microcapsule average size notably increased, most likely due to coalescence of the formed microcapsules.

Interestingly, size distribution analysis of DLS experiments at different pHs (Appendix A) showed a single size distribution at pH 7.3 (isoelectric point) and at pH 5.5 (preparation condition), while at acid (pH 4.0 and pH 1.9) and alkaline (pH 8.3 and pH 11.2) pHs, two and three size distributions were obtained by deconvoluting DLS data. These results clearly highlight the heterogeneous dimensions of ZP/SO MCs at those extreme pH values.

Zeta potential (ζ) measurements revealed that, at pH 5.5 (preparation condition), ζ = +35.4 mV, a clear indication of the stability of ZP/SO MCs under these experimental conditions. In this case, repulsive interactions between the charged ZP interfaces inhibited the growth of protein microcapsules by coalescence. Interestingly, at the isoelectric point (pH = 7.3), the zeta potential markedly decreased (ζ = +18.6 mV), suggesting that at this pH, coalescence phenomena could be favored.

### 2.5. Encapsulation and Release of Curcumin into/from ZP/SO Microcapsules

Curcumin [bis(4-hydroxy-3-methoxyphenyl)-1,6-heptadiene-3,5-dione] is a natural yellow-orange dye derived from the rhizome of *Curcuma longa*, a member of the ginger family Zingiberaceae. It shows unique pharmaceutical and nutraceutical properties due to its antioxidant, antibacterial, and anti-inflammatory activities, as well as recognized safety [52].

Besides its relevant bioactive properties, curcumin has been chosen for inclusion studies because it lacks solubility in water, is soluble in oil up to 2.90 ± 0.15 mg per oil gram [53], and is strongly fluorescent [54]. These properties make curcumin a proper candidate for its successful encapsulation in o/w microcapsules and a suitable probe for fluorescence imaging. The inclusion of curcumin in ZP–polymer nanoparticles [55,56,57,58,59,60,61,62,63] and hybrid assemblies [64] has already been studied for the delivery of food ingredients or therapeutics. In this study, curcumin was embedded in the oily inner phase of pure ZP microcapsules prepared by UAE.

The UV-Vis absorption, fluorescence emission, and excitation spectra of curcumin in SO and EtOH are reported in Appendix A, respectively. As can be seen in the reported figures, the absorption and fluorescence spectra of curcumin in SO are more structured with respect to the absorption and emission spectra in EtOH. In particular, the fluorescence emission peak of curcumin is shifted from λ_max_ = 486 nm in SO to λ_max_ = 530 nm in EtOH as a consequence of spectral relaxation in a polar solvent [65].

Curcumin has therefore been used as a probe to test the ability of ZP/SO MCs to encapsulate an active ingredient using UAE. Bright-field optical and fluorescence microscopy measurements revealed that curcumin was successfully encapsulated into the ZP/SO MCs oily phase, as shown in Figure 9. It should be noted that the curcumin fluorescence signal (Figure 9A) overlaps with the ZP/SO microcapsules imaged by bright-field optical microscopy (Figure 9B), confirming the successful inclusion of curcumin in the inner oily phase of ZP/SO MCs.

Notably, the fluorescence emission spectrum of curcumin in ZP/SO MCs (λ_em,max_ = 498 nm) occurred in a wavelength region intermediate between those of its fluorescence emission spectra in SO (λ_em,max_ = 486 nm) and EtOH/H_2_O 70/30 (λ_em,max_ = 530 nm), as shown in Appendix A. This result indicates that curcumin was encapsulated in the oily phase of ZP/SO MCs, but it was experiencing a more polar environment [66]. This finding suggests a preferential location of curcumin in the ZP inner periphery or in the ZP coacervates. This hypothesis is supported by the results of Hu et al., who synthesized core-shell ZP nanoparticles coated by a hydrophilic pectin outer layer fortified in the hydrophobic inner phase by curcumin. Interestingly, curcumin was found to interact preferentially with hydrophobic ZP areas through its aromatic groups and inter-ring chains [55].

CLSFM imaging of ZP/SO/curcumin MCs was carried out, measuring the curcumin fluorescence emission when excited at λ_ex_ = 418 nm. Figure 10 clearly shows that curcumin was homogeneously embedded into the microcapsule inner phase and that ZP/SO/coumarin MCs maintained their spherical morphology upon the curcumin inclusion. Interestingly, CLSFM measurements also revealed the presence of low-fluorescent curcumin coacervates in water solution and in the inner core of ZP/SO MCs (Figure 10).

These findings allowed us to use the very different fluorescence quantum yield of curcumin in SO and in EtOH/H_2_O to monitor the curcumin release from ZP/SO MC upon pH and temperature stimuli.

Interestingly, by varying the pH from pH 5.5 to pH 1.7, a marked decrease in the curcumin fluorescence intensity was observed (Appendix A). This finding indicates that the critical pH for curcumin release from ZP/SO MCs is around pH = 4, as also suggested by pH-dependent DLS experiments (Appendix A).

Temperature-dependent RLS and DLS experiments also showed that a significant reduction in the number density and size of ZP/SO MCs could be detected between 30° and 40 °C (Appendix A, respectively).

Therefore, to obtain information on the mechanism of curcumin release from ZP/SO MCs, we analyzed the time dependence of the curcumin fluorescence intensity in this temperature range. In particular, kinetic experiments, carried out at T = 32, 35, 37, and 40 °C on ZP/SO/curcumin MCs at pH 5.3, revealed an exponential decrease in the curcumin fluorescence intensity as a consequence of the curcumin diffusion from the core of ZP/SO MCs to the EtOH/H_2_O solution (Figure 11).

The fractional and total curcumin released in solution, calculated from the data reported in Figure 11 at each temperature, are reported in Table 2.

A biexponential fitting of the curcumin fluorescence emission intensity decays with time was carried out through the equationF(t) = α_1_ exp(−k_1_t) + α_2_ exp(−k_2_t)
and provided the temperature-dependent rate constants reported in Table 3.

These results indicate that the curcumin release from ZP/SO MCs follows a complex mechanism, characterized by two kinetic processes differing by a one-order-of-magnitude time scale.

The curcumin fluorescence intensity decays were also fitted using a logistic Weibull model, suitably adapted to a dissolution/delivery process (Appendix A, Appendix A) [67]. Interestingly, a shape parameter (b) consistently less than 1 was obtained at all temperatures (b = 0.61 ± 0.04). This value is typical of a site-specific biphasic release kinetics, in which the drug diffusion is coupled to partial erosion of the protein shell [68,69]. These findings strongly suggest that, in our case, the curcumin diffusion out of the oily core of ZP/SO MCs is enhanced at higher temperatures by the collapse of MCs reaching a critical size (thermal disruption), in agreement with RLS and DLS findings.

## 3. Materials and Methods

### 3.1. Materials

#### 3.1.1. Products

ZP was supplied by Sigma Aldrich (Milan, Italy) [maize z1C2 (541924, UniProt access numbers Q41896 and P04700)] as dry powder and used as such without further purification or pretreatment. According to the product information, the sample consists of two αZP of 22 and 24 kDa, characterized by a 66% and 63% helical conformation, respectively. ZP was solubilized in EtOH/H_2_O 70/30 (*v*/*v*) under stirring. All the solutions for spectroscopy and microscopy experiments were freshly prepared from this same batch and sealed in a dry, cool, and dark environment.

Curcumin (powder, min. 95% p/p) was purchased from Galeano srl (Rome, Italy) and used as such without further purification. SO, NR, and RhB were purchased by Sigma Aldrich. Spectroscopic absolute EtOH (anhydrous, ≥99.9%) was purchased from Carlo Erba (Milan, Italy). Milli-Q filtered water was obtained from a Millipore system (Merck Millipore, Milan, Italy) (18.2 MΩ cm) at 25 °C.

#### 3.1.2. Preparation of o/w ZP/SO Microcapsules

Optimal experimental conditions for the preparation of o/w ZP/SO MCs by UAE were established by carrying out several experiments at different ZP concentrations (5, 7.5, 10 mg/mL), soybean oil volumes (10, 20, 50 μL), ultrasound applied powers (110, 165, 220 W), and sonication times (25, 30, 45 s) (Appendix A). In a typical experiment, 10 μL SO was added to 5, 7.5, or 10 mg ZP dissolved in 1 mL EtOH/H_2_O 70/30 (*v*/*v*). A 3 mm diameter ultrasound horn (Branson Digital Sonifier, ThermoFisher Scientific, Ferentino, Italy) was placed at the o/w interface. Sonication was carried out at 20 kHz frequency for 25 s at 220 W acoustic power. To reduce the temperature shock at the o/w interface, the system was immersed in an ice bath during the sonication process. The obtained microcapsules were stored in a test tube at room temperature. Long-time storage of ZP/SO MCs was carried out at 4 °C in a dry environment and in the dark. All experiments were carried out in triplicate.

#### 3.1.3. Inclusion/Release of Dyes and Active Compounds in ZP/SO Microcapsules

NR was included in ZP/SO microcapsules using SO stained with NR during the microcapsule preparation. The stained SO was stored under dark conditions at 4 °C. NR UV-Vis absorption and fluorescence spectra in SO peaked at λ_max_ = 510 nm and λ_max_ = 570 nm, respectively.

ZP/SO MCs were also stained by dissolving RhB in ZP EtOH/H_2_O 70/30 (*v*/*v*), following the standard conditions for microcapsule preparation. RhB UV-Vis absorption and fluorescence spectra are located in the 500–580 nm and 550–680 nm wavelength regions, respectively.

Next, 1.7 mg of curcumin (bis(4-hydroxy-3-methoxyphenyl)-1,6-heptadiene-3,5-dione) was added to 1 mg of SO and dissolved under stirring, with the curcumin/SO solution stored under dark conditions at 4 °C. Curcumin/SO was included in ZP/SO MCs under standard operative conditions for microcapsule preparation. The release of curcumin from ZP/SO MCs under temperature and pH stimuli was monitored by measuring the decrease in curcumin fluorescence emission intensity when the drug, embedded into the SO inner phase, entered the aqueous phase.

### 3.2. Methods and Instrumentation

#### 3.2.1. UV–Vis Absorption Spectroscopy

UV-Vis absorption experiments were carried out with a Cary 100 Scan spectrophotometer (Varian, Middelburg, The Netherlands) equipped with a Peltier thermostat (JASCO EHCS-760, JASCO Europe, Cremella, LC, Italy), using quartz cuvettes (Hellma Analytics, Mulheim, Germany) with optical lengths of 1 and 0.5 cm. The EtOH/H_2_O 70/30 (*v*/*v*) spectrum was subtracted for background correction in all the optical spectroscopy experiments. All the spectroscopic experiments were carried out in triplicate.

#### 3.2.2. Electronic Circular Dichroism (ECD)

ECD experiments were carried out using a Jasco J-1500 CD spectropolarimeter (JASCO Europe) using a cell holder equipped with a Peltier thermostat (PTC-510, JASCO Europe). The setup was purged with ultra-pure nitrogen gas. Spectra were recorded in the far-UV wavelength range, i.e., from 190 to 250 nm (amide region) and from 250 to 350 nm for the aromatic region (tyrosine), using a scan speed of 20 nm/min, a bandwidth of 2 nm, and a sensitivity of 20 mdeg. For each sample, the spectra were accumulated 10 times to maximize the signal-to-noise ratio. The solvent spectrum was subtracted for background correction. Quartz cuvettes (Hellma Analytics) with optical lengths of 0.1 and 1 cm were used for CD measurements in the amide and aromatic regions, respectively. Data analysis for secondary structure determination was performed relying on the BestSel code [50].

#### 3.2.3. Steady-State Fluorescence

Steady-state fluorescence experiments were carried out using a Fluoromax-4 spectrofluorometer (Horiba, Jobin Yvon, IBH Glasgow, UK) equipped with automatically controlled Glan–Thomson polarizers and single photon counting (SPC) detection. The temperature was set through an external thermostat, Julabo F25 (Sigma-Aldrich), allowing for a maximum oscillation of ±0.1 °C. Emission and excitation spectra were acquired using a 1 cm quartz cell (Hellma Analytics) and 3/3 nm excitation (ex) and emission (em) slit openings. Then, a 4/10 mm asymmetric quartz cell (Hellma Analytics) and 1/1 nm ex/em slits were used for curcumin in SO, while a 5 mm quartz cell (Hellma Analytics) and 1.5/1.5 nm ex/em slits were used for spectral measurements of curcumin embedded in ZP/SO microcapsules. For this experiment, a cut-off emission filter was used to minimize light scattering contamination. Unless otherwise stated, fluorescence intensities are reported as signal-to-reference [count per second (cps)/microamperes (µA)] units. All the measurements were carried out with optical densities lower than 0.1 at the excitation wavelength to minimize inner filter effects (IFEs). IFE correction was applied to all the measured fluorescence spectra [66].

#### 3.2.4. Optical Microscopy

Widefield microscopy images were obtained with a Zeiss microscope, Axio Scope A1 (Carl Zeiss, Oberkochen, Germany), equipped with a Mercury Lamp HBO 50 (Osram, Milan, Italy) and CCD AxioCam ICm1 (Zeiss Italia, Milan, Italy). The images were acquired with the software Zen-Blue (version 2.0.0.0) (Zeiss Microscope, Oberkochen, Germany) using magnification air (10×, 20×, 40×, and 63×) and oil immersion (100×) objectives. Fluorescence imaging was performed in a single-channel mode of excitation (red channel for NR and RhB, blue channel for curcumin). Images and particle diameters were analyzed using the software ImageJ (version 1.54g) [70].

Confocal laser scanning fluorescence microscopy experiments were carried out using an Olympus IX-81 inverted microscope (Olympus Optical Co., Hamburg, Germany) coupled to a confocal scanning microscope FV1000 (Olympus, Shinjuku, Tokyo, Japan). Laser excitation was achieved at λ_exc_ = 495 and 514 nm. Images were captured with a 40× oil immersion objective, adding 4× and 5× zoom (160–200× total magnification). Three-dimensional image reconstruction was developed using commercial software.

#### 3.2.5. Field Emission Environmental Scanning Electron Microscope (FE-ESEM)

ZP/SO MCs were imaged by an FE-ESEM LEO 1530 (Zeiss Microscope) microscope, applying a beam accelerating voltage of 3 kV. Measurements were carried out by placing a 5 μL ZP/SO MCs solution on a graphite substrate, vacuum-packed for 30 min before recording topography information [71].

#### 3.2.6. Light Scattering Experiments

Rayleigh light scattering (RLS) intensities of ZP/SO MCs solutions were measured using the same fluorescence equipment described above, reporting the RLS intensities as signal-to-reference [count per second (cps)/microamperes (µA)]. Both excitation and emission monochromators were set at λ_ex_ = λ_em_ = 600 nm, well outside the sample absorption or fluorescence emission regions. Notably, 5 × 5 mm quartz cells (Hellma Analytics) were used, with 0.5/0.5 nm ex/em slit openings.

Dynamic light scattering (DLS) experiments were carried out at room temperature using a DLS Zetasizer Nano ZS (ZEN0040, Malvern Instruments, Malvern, UK) equipped with a He-Ne laser. ζ-potential measurements were carried out by the same equipment using appropriate cells. All the experiments were carried out in triplicate.

## 4. Conclusions

In this work, ZP/SO MCs were prepared by UAE under optimal experimental conditions. It was shown that the size distribution and number density of ZP/SO MCs were controlled by the combined effects of sonication experimental conditions and ZP/SO concentration ratio. In particular, ZP concentration turned out to be the most influential parameter with regard to the particle number density and size, while the applied acoustic power and sonication time mainly affected the microcapsule structure and integrity over time. Combining all the available information, optimal experimental conditions for preparing stable o/w ZP/SO MCs were determined as follows: 1/20 oil/[EtOH/H_2_O 70/30 (*v*/*v*)] ratio, 10 mg/mL ZP concentration, 10 µL SO, 20 kHz sonication frequency, 220 W acoustic power, 25 s sonication time in an ice bath. Under these experimental conditions, ZP/SO MCs with average diameters of 0.9 ± 0.3 μm, showing reduced MC coalescence and particle number density drop over 5 days, were obtained.

The ZP/SO MCs morphology was characterized by optical microscopy, CLSFM, and FE-ESEM imaging, providing direct evidence of the structure of protein microcapsules. It was observed that the ZP-coated microcapsules grew until a critical size was achieved, followed by collapse and rupture of the protein microcapsules. Dispersibility and stability in water of dried ZP/SO MCs were also investigated. The results showed that, under appropriate storage conditions, ZP/SO MCs can be successfully transferred to and stored in 100% aqueous solutions. We consider this result to be of crucial relevance for future applications.

Fluorescence spectroscopy experiments showed that ZP/SO MCs are stabilized by the formation of tyrosine–tyrosine cross-linking and interprotein stacking interactions. CD data supported this scenario, as they displayed a protein structural reorganization from α-helical to antiparallel β-sheet structures.

The stability of ZP/SO MCs in the 20–70 °C temperature range and 2–11 pH interval was investigated by DLS experiments. It was found that the size of the protein microcapsules decreased almost continuously with increasing temperatures and at lower pHs, while at alkaline pHs, coalescence of the microcapsules, followed by their rupture and formation of micrometric oil droplets, was observed.

Finally, as a proof-of-principle experiment, the ability of ZP/SO MCs to encapsulate curcumin, a fluorescent molecule showing unique pharmaceutical and nutraceutical properties, was tested and confirmed by optical microscopy and LCSFM imaging. The release of curcumin from ZP/SO MCs upon pH and temperature stimuli was also investigated by monitoring the decrease in the curcumin fluorescence intensity upon diffusion from the microcapsule inner oily phase to the aqueous environment. Kinetic experiments in the critical temperature region between 32 and 40 °C revealed the complex character of the curcumin diffusion pathways, most likely associated with the coupling of the drug diffusion process and protein shell erosion.

It should be noted that the utilization of curcumin in food and supplement products is still limited because of its extremely low (3 × 10^−8^ M) water solubility, poor chemical stability, and low oral bioavailability under physiological conditions. These drawbacks make curcumin difficult to incorporate into many products and to solubilize in the aqueous fluids within the gastrointestinal tract. Encapsulation seems to be the method of choice to enhance curcumin solubility, stability, and bioavailability [55,56,57,58,59,60,61,62,63,64,72].

The optimized UAE encapsulation protocol proved to be successful, and curcumin was efficiently included and retained in ZP/SO MCs. These results set promising ground for future studies on the subject, considering the existing barriers to ZP transfer and delivery [73,74]. In vivo studies concerning the capacity of ZP microcapsules to preserve the encapsulated curcumin and allow for its controlled release in specific application environments will be the object of future studies.

## Figures and Tables

**Figure 1 molecules-31-00153-f001:**
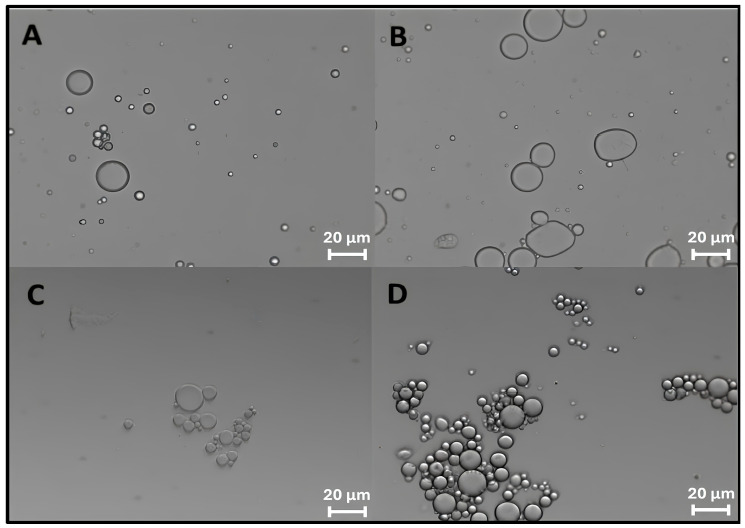
Optical microscopy images of o/w emulsions from 1 mL EtOH/H_2_O 70/30 (*v*/*v*) ZP solution and 50 µL soybean oil: (**A**) 2.5 mg/mL ZP, t = 0; (**B**) 5 mg/mL ZP, t = 0; (**C**) 2.5 mg/mL ZP, t = 4 days; (**D**) 5 mg/mL ZP, t = 4 days.

**Figure 2 molecules-31-00153-f002:**
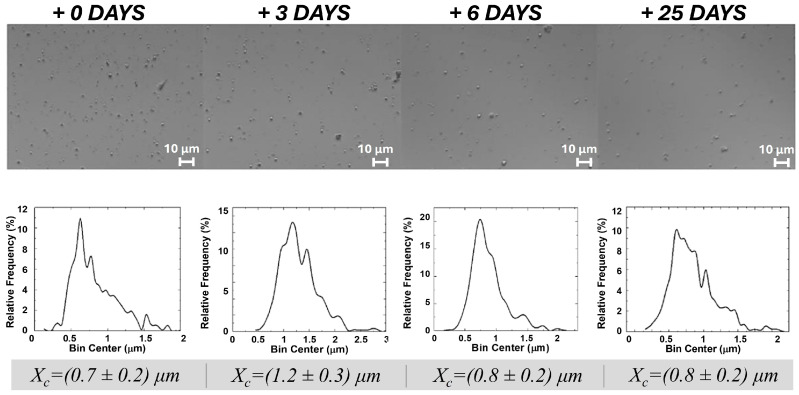
Optical microscopy images and size distribution of dried o/w ZP/SO MCs in 3 mL H_2_O from the day of redispersion (0) to day 25. The test tube was gently shaken before each sampling. Preparation: UAE frequency = 20 kHz, power = 220 W, sonication time = 25 s, continuous mode; 10 mg/mL ZP, 10 µL SO.

**Figure 3 molecules-31-00153-f003:**
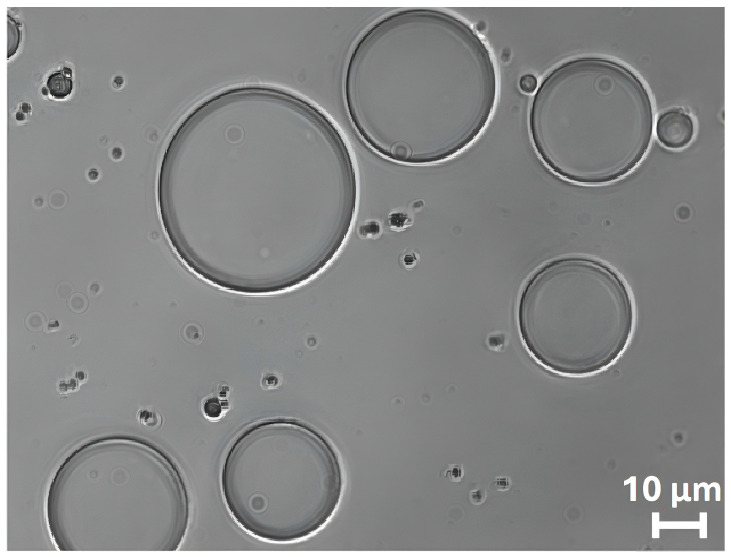
Bright-field optical microscopy image (63×) of o/w ZP/SO microcapsules prepared by UAE of 1 mL 5.0 mg/mL ZP EtOH/H_2_O 70/30 (*v*/*v*) and 10 µL soybean oil solution. Sonication conditions: frequency = 20 kHz; power = 220 W; continuous mode.

**Figure 4 molecules-31-00153-f004:**
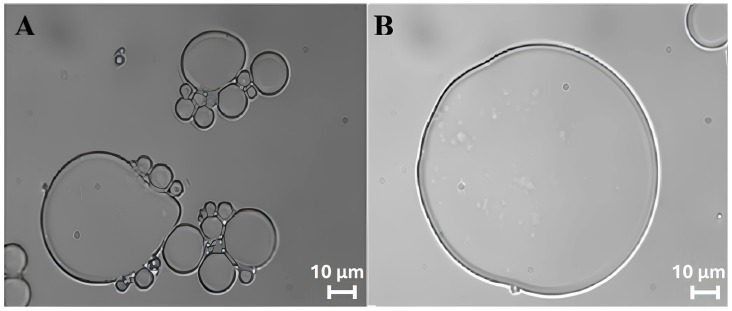
Bright-field microscopy images (63×) of o/w ZP/SO microcapsules 4 days after UAE of 1 mL of 5.0 mg/mL ZP EtOH/H_2_O 70/30 (*v*/*v*) solution and 10 µL soybean oil. Sonication conditions: frequency = 20 kHz; power = 220 W; continuous mode. (**A**): ZP/SO microcapsules undergoing coalescence; (**B**): a ‘giant’ ZP/SO microcapsule.

**Figure 5 molecules-31-00153-f005:**
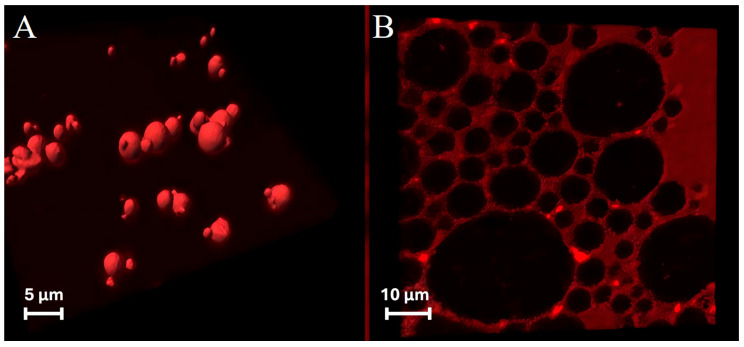
Three-dimensional fluorescence confocal microscopy image of o/w ZP/SO microcapsules obtained by UAE of a 10 mg/mL ZP EtOH/H_2_O 70/30 (*v*/*v*) and 10 μL soybean oil solution: (**A**) Nile red added to the SO before the synthesis; (**B**) rhodamine B added to the ZP solution before the synthesis. Sample volume: 1 mL; sonication conditions: power = 220 W; time = 45 s, frequency = 20 kHz; continuous mode; test tube in ice bath.

**Figure 6 molecules-31-00153-f006:**
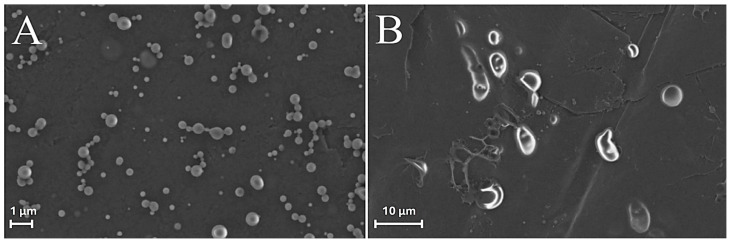
FE-ESEM images of o/w ZP/SO emulsions from 10 mg/mL zein EtOH/H_2_O 70/30 (*v*/*v*) and 10 μL soybean oil solutions: (**A**) day of synthesis. (**B**) after 18 days. Sonication parameters: power = 220 W; time = 25 s; frequency = 20 kHz; continuous mode.

**Figure 7 molecules-31-00153-f007:**
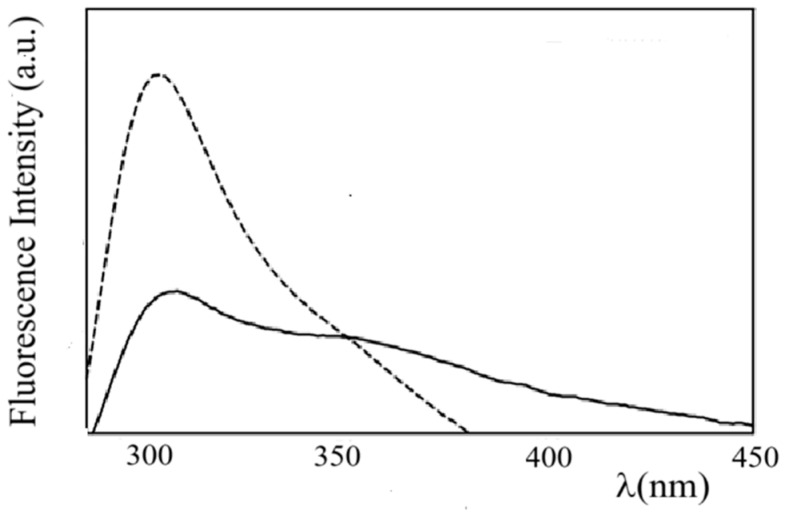
Fluorescence emission spectra of zein protein in EtOH/H_2_O 70/30 (*v*/*v*) (dashed line) and ZP/SO microcapsules (continuous line) (λ_ex_ = 278 nm).

**Figure 8 molecules-31-00153-f008:**
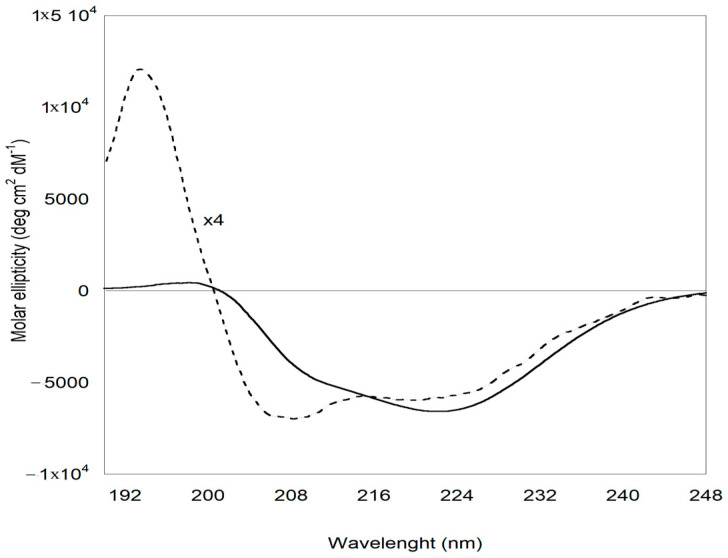
Electronic circular dichroism spectra of zein protein in EtOH/H_2_O 70/30 (*v*/*v*) (dotted line) and ZP/SO microcapsules (continuous line). The CD profile of ZP is magnified by a factor of 4 for clarity.

**Figure 9 molecules-31-00153-f009:**
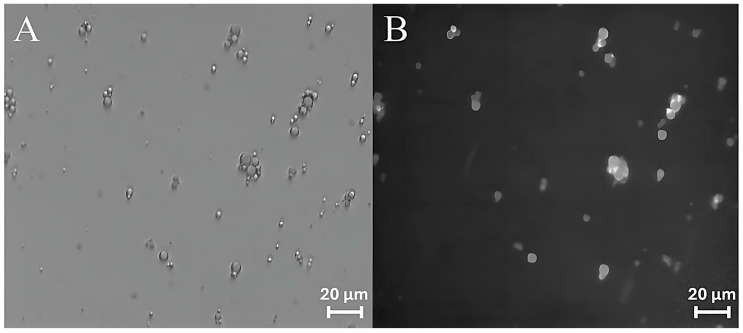
Bright-field optical (**A**) and fluorescence (**B**) microscopy images of ZP/SO microcapsules encapsulating curcumin. Curcumin was included in soybean oil prior to sonication.

**Figure 10 molecules-31-00153-f010:**
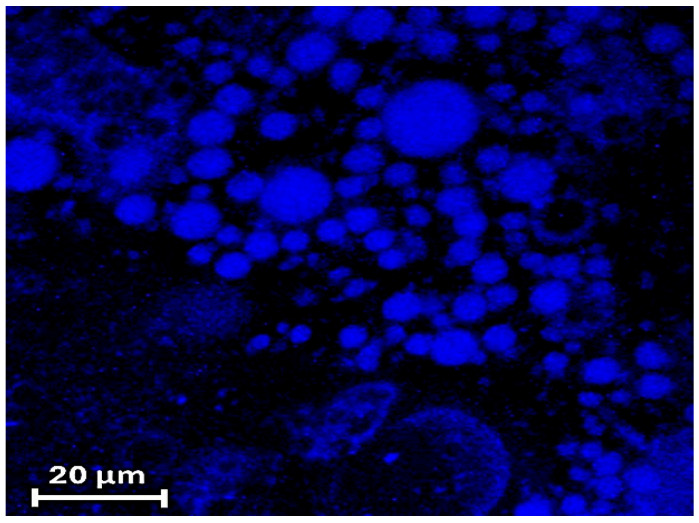
Confocal laser scanning fluorescence microscopy imaging of ZP/SO/curcumin microcapsules. Note the presence of low-fluorescent ZP/curcumin coacervates in aqueous solution and embedded in the inner oily phase of ZP/SO microcapsules.

**Figure 11 molecules-31-00153-f011:**
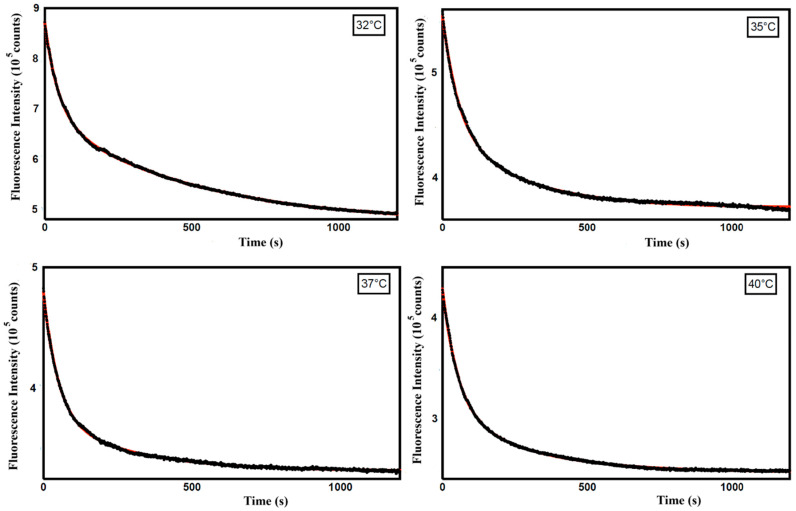
Fluorescence emission intensity as a function of time (seconds) of curcumin in ZP/SO/curcumin microcapsules at different temperatures (32 °C, 35 °C, 37 °C, and 40 °C). Black line: experimental data; red line: biexponential fitting curve.

**Table 1 molecules-31-00153-t001:** Secondary structure of ZP in solution and in ZP/SO microcapsules from ECD experiments (T = 25 °C).

Δ(%)	ZP (ZP/SO Microcapsules) (%)	ZP (70/30 *v*/*v* EtOH/H_2_O) (%)	Secondary Structure
−34.3	14.3	48.6	α-helix
36.3	42.4	6.1	antiparallel β-sheet
−0.1	13.8	13.9	β turns
−1.7°	29.6	31.3	others

**Table 2 molecules-31-00153-t002:** Curcumin fractional and total release at each temperature from fluorescence intensity measurements.

Temperature (°C)	Total Release (%)	Fractional Release at Each Temperature (%)
32	43.6	43.6
35	33.9	57.7
37	31.4	62.1
40	41.8	71.5

**Table 3 molecules-31-00153-t003:** Kinetic rate constants (k_i_) and weights (a_i_) obtained from a biexponential fitting of the time-dependent curcumin fluorescence emission intensity at different temperatures (pH 5.3).

Temperature (°C)	k_1_ (10^−2^ s^−1^) (a_1_)	k_2_ (10^−3^ s^−1^) (a_2_)
32	2.14 ± 0.02 (0.47)	2.00 ± 0.02 (0.53)
35	1.81 ± 0.05 (0.58)	3.8 ± 0.1 (0.42)
37	1.88 ± 0.03 (0.69)	3.53 ± 0.08 (0.31)
40	1.93 ± 0.02 (0.64)	3.66 ± 0.05 (0.36)

## Data Availability

The data presented in this study are available on request from the corresponding author.

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
