# Peer review of "The Remarkable Properties of Oil-in-Water Zein Protein Microcapsules"

_molecules, 2026, doi:10.3390/molecules31010153_

Round 1

Reviewer 1 Report

Comments and Suggestions for Authors

Dear Authors,

  1. The tittle of your manuscript is not understandable. „Oil-in-Water Zein Protein/Soybean Oil Microcapsules…” - please, simplify the tittle.
  2. Abstract: you wrote „oil in water (o/w) solutions” – o/w it is an emulsion/dispersion not solution
  3. Introduction: Please, add the information what is novel in your experiment. How does it differ from other manuscripts?
  4. ” ZP/SO MC” – what is MC? Microcapsules? There is no explanation of this abbreviation
  5. Materials and methods: line 400 „were added to 5 to 10 mg” or rather should be „from 5 to 10 mg”?
  6. Part UV–Vis absorption spectroscopy – what was the blank?
  7. Materials and methods - How many repetitions were the tests performer? Lack of this information in the description of the methods and in the description of figures and the tables n=3 or n=4?
  8. References: For some references, when you press the doi number, the following message appears: doi not found (e.g. reference no 68)

Author Response

Thanks for your careful work of revision, taht allowed us to improve the quality of the paper.

See the attached file for reply to your comments.

Best regards

Mariano Venanzi

Reviewer 2 Report

Comments and Suggestions for Authors

I recommend major revision for this work. Some more comments below

Novelty has to be better articulated, its doubtful in the way how its written. What is original in Authors work in comparison to previous studies?

Saying Zein has amphiphilic and has self-assembling properties is too vague.  Authors should carry out experiments to demonstrate this convincingly. Authors also call Zein hydrophobic in Introduction which contradicts with the Authors main claim. So, the Zein is amphiphilic or still hydrophobic? I do believe Zein is hydrophobic. So very little quantities are available to participate into interactions Authors describe.

How can you see that microcapsules on Figure 1 contain Zein?

Regarding comments on Figure 2. Morphology is not globular, its spherical.

Regarding comment on Figure 4. I do not see convincing arguments for Zein presence in microcapsule shell.

Curcumin degrades in water rapidly at different temperature and pH, was this taken into account during the experiments?

I believe that Materials and Methods should appear in the manuscript first, and Results afterwards?

Conclusions need to be shortened and written in more focused way.

Author Response

Dear reviewer,

thanks for your careful work of revision that allowed us to improve the quality of the paper.

Reply to your comments are reported in the attached file.

best regards

Mariano Venanzi

Reviewer 3 Report

Comments and Suggestions for Authors

The manuscript "Oil-in-Water Zein Protein/Soybean Oil Microcapsules from Ultrasound-Assisted Emulsification: Stability, Morphology and Curcumin Inclusion/Release Studies" aims to demonstrate the production of zein-based microcapsules and to evaluate their stability, morphology, and curcumin inclusion/release. The topic is relevant and related to the development of biopolymer-based capsule applications. The development of biopolymer based systems is a rapidly growing field. However, several parts could be improved to increase the scientific impact, the readability, and the robustness of the work.

  1. Please broaden the novelty statement at the end of the introduction.
    The current References (lines 60–90) include common aspects of zein and ultrasonic emulsification but do not clearly separate this work from previous studies, e.g., Hu, K. et al., Food Hydrocolloids, 2018, 77, 607–616 (zein-curcumin nanoparticles) and Hu, K. et al., Food Hydrocolloids, 2018, 77, 607–616 (zeino-curcumin nanoparticles) and Leong, T.S.H. et al., Ultrasonics Sonochemistry, 2017, 35, 605–613 (ultrasonic emulsions).

Please clearly identify what is new here, e.g., improved long-term stability, re-dispersibility, or microcapsule sensitivity to pH.

  1. In paragraph 2.2 (Preparation of O/W ZP/SO microcapsules) and in the supplementary table SMT1, please specify:
    The number of repeats (n) for each experimental condition.

Do the particle size data represent the mean ± SD or a single measurement?
The type of DLS instrument used and the measurement temperature.

When you describe the curcumin insertion experiment (Section 2.4), please indicate the exact concentration of curcumin in soybean oil before the emulsification and whether the capsule's efficacy was determined quantitatively or qualitatively.

  1. Results and Discussion

1.Effect of Sonication Power and Duration (Fig. 1–2, SMF1–SMF2)

-The discussion (lines ~150–200) is mostly of a descriptive nature. Please include a quantitative evaluation of the effect of power and sonication duration on particle sizes (e.g., linear or logarithmic correlation).

-Please clarify whether a temperature increase was observed during 45 seconds of sonication, as this may partially degrade zein and affect droplet stabilization.

  1. Effect of protein concentration (Fig. 3, SMF3)

-Please provide numerical values for the average particle size at each protein concentration (e.g., 5, 7.5, 10 mg/mL) and its changes over time (0–11 days).

- Please include error bars or mention the variability of the data.

  1. Storage studies (SMF4)

-The statement that "dry storage in vacuum is favorable for integrity" (SM1-Storage) should be based on quantitative evidence (e.g., comparison of DLS data before and after redispersion).

  1. Morphological characterization (SMF7–SMF9)

- Figures SMF7 and SMF8 (fluorescence microscopy) are informed, but are missing visible scale bars. Please improve the resolution of the images and ensure that the scale bars are clearly marked.
-Please include a brief note on whether the distribution of fluorescence intensity shows homogeneous encapsulation of curcumin.

5.Optical spectroscopy and stability (Figs. 4–6, SMF10–SMF14)
-Please describe the solvent background correction technique used for UV-Vis and fluorescence measurements.

- Please add a paragraph discussing the changes induced by temperature and pH in the context of the secondary structure of zein (formation of β-sheets and loss of α-helices). The references to CD spectroscopy literature (e.g., Wang et al., J. Agric. Food Chem., 2019) would strengthen the argument.

  1. Curcumin incorporation and release (Figs. 7–8, SMF15–SMF18)

-The dissolution profiles should be reported in quantitative terms. Please specify:

-The release percentage of curcumin at a specified time (e.g., after 1 h, 6 h, 24 h).

-The total encapsulation efficiency (EE%) and the fit model used (first-order, double exponential, or Korsmeyer–Peppas).

-Please compare your results with other zein-based systems described in the literature to contextualize your findings (e.g., Luo et al., Colloids Surf. B, 2020).

  1. for figures (1,3 and 5) and tables, increase the resolution and font size.

- Increase the resolution and font size. Ensure that all axes and labels are clear.

- Table 1 (ECD data): add a column showing the Δ% change between samples (e.g., α-helix and β-sheet content).

- Supplementary figures (SMF5–SMF18): clearly identify all references to figures in the main text.

Comments on the Quality of English Language

The language is clear and understandable, and scientific terminology is used correctly. Several sentences are too long, though, and could be rewritten to make the text easier to read. Minor grammatical and typo-graphical corrections are suggested, in particular concerning the use of the articles, the consistency of verb tenses, and the spacing between units (e.g., "µL" and "uL"). In general, the manuscript would be improved by careful editing to make the text more fluent and accurate.

Author Response

Dear Reviewer,

thanks for your careful work of revision, that allowed us to improve the quality of the paper.

In the attached file you can find our reply to your comments and requests of revision.

best regards

Mariano Venanzi

Round 2

Reviewer 2 Report

Comments and Suggestions for Authors

comments are addressed

Reviewer 3 Report

Comments and Suggestions for Authors

The manuscript has been thoroughly revised and significantly improved. The authors have adequately addressed the main issues raised during the review process, including explanation of novelty, revision and strengthening of results and supplementary materials. Overall, the manuscript meets the scientific standards of the journal. Based on the present version, I consider the manuscript suitable for publication.